Structure and pigment make the eyed elater’s eyespots black

Wong Victoria L. 1 2
Marek Paul E. pmarek@vt.edu 1
1 Department of Entomology, Virginia Polytechnic Institute and State University (Virginia Tech) , Blacksburg , VA , United States of America
2 Department of Entomology, Texas A&M University , College Station , TX , United States of America
Osborn Karen
Electronic publication date: 2020 Jan 13
Publication date: 2020
Volume: 8
Electronic Location ID: e8161
Received 2019 Jun 17; Accepted 2019 Nov 4
Copyright year: 2020
License: This is an open access article, free of all copyright, made available under the Creative Commons Public Domain Dedication. This work may be freely reproduced, distributed, transmitted, modified, built upon, or otherwise used by anyone for any lawful purpose.
License URL: https://creativecommons.org/publicdomain/zero/1.0/

Keywords: Super black, Melanin, Beetle, Deimatic, Aposematic, Color, Spectrum, Scattering, Startle, Eyespot

Funding: National Science Foundation DBI # 1458045 USDA NIFA Hatch Project VA-160028 This work was supported by the National Science Foundation (DBI # 1458045), and by Virginia Tech, Department of Entomology, College of Life Sciences, and by a USDA NIFA Hatch Project (VA-160028). The funders had no role in study design, data collection and analysis, decision to publish, or preparation of the manuscript.

==============================
Surface structures that trap light leading to near complete structural absorption creates an appearance of “super black.” Well known in the natural world from bird feathers and butterfly scales, super black has evolved independently from various anatomical structures. Due to an exceptional ability to reduce specular reflection, these biological materials have garnered interest from optical industries. Here we describe the false eyes of the eyed elater click beetle, which, while not classified as super black, still attains near complete absorption of light partly due to an array of vertically-aligned microtubules. These cone-shaped microtubules are modified hairs (setae) that are localized to eyespots on the dorsum of the beetle, and absorb 96.1% of incident light (at a 24.8° collection angle) in the spectrum between 300–700 nm. Filled with melanin, the setae combine structure and pigment to generate multiple reflections and refractions causing light to travel a greater distance. This light-capturing architecture leaves little light available to receivers and the false eyes appear as deep black making them appear more conspicuous to predators.

Introduction

Black in nature is often achieved by pigments (e.g., melanin) that absorb most visible light (Zhang et al., 2017; Hsiung, Blackledge & Shawkey, 2015). In some cases, only ultraviolet light (320–400 nm) is reflected, such as in Asian whistling-thrushes (Andersson, 1996). Often, black pigment is overlaid by a glossy surface thereby imparting specular reflection increasing at angles normal to the illumination source, for example in many beetles (Seago et al., 2009). Among insects, black pigmentation is typically achieved during the process of molting and tanning including sclerotization and melanization. In contrast, super black in butterflies, birds, and snakes is usually achieved by structural absorption (Vukusic, 2009) of nearly all (≥99%) light. Three-dimensional structures, such as forest-like arrays of microtubules or polydisperse honeycomb-like meshes on butterfly wings and the highly ramified barbules on bird of paradise feathers, act as a baffle to light (Vukusic, Sambles & Lawrence, 2004; Han et al., 2015; McCoy et al., 2018). (But, there are instances of “pseudo”-black achieved through additive mixing of structural green and magenta iridescence Seago et al., 2009). In some of these instances, super black structures evolved to impart varying degrees of reflection (blackness) dependent upon angle (Han et al., 2015). In others, structural absorption is assisted by melanin, and in jumping spiders, brush-like 3D scales absorb light and stray light is recaptured by an underlying melanin-containing layer (McCoy et al., 2019). Functional explanations of the evolutionary origins of super black include sexual selection (McCoy et al., 2018), predator defense (Spinner et al., 2013), and hydrophobicity (Maurer, Kohl & Gebhardt, 2017). In most of these examples of structural black, there is an array of protuberances on the surface of the animal that are perpendicularly oriented. These forest-like arrays of protuberances, which vary in composition from setae, microtrichia, barbules, and scales, have evolved repeatedly across the Tree of Life. Super black surface structures from nature have been applied to human industry since they have application for solar technology, a coating for the internal barrels of lenses in optical manufacturing, and artistic expression  (McCoy et al., 2019; Kennedy, 2016; Zhao et al., 2011).

Known colloquially as the eyed elater or Eastern eyed click beetle, Alaus oculatus (Linneaus, 1758) (Figs. 1A, 1B) is a common beetle in the eastern US with large and conspicuous eye-like spots on its back (McDermott, 1911). False eyes have evolved independently in several lineages of insects including moths, cockroaches and mantises, butterflies, and beetles (Misof et al., 2014). Two lineages of click beetles (family Elateridae) in the subfamily Agrypninae possess ostensibly false eyes, including some members of the genus Alaus and individuals of the tribe Pyrophorini (Kundrata, Bocakova & Bocak, 2014). The latter possess bioluminescent eyespots atop the pronotum and include the headlight elater (Pyrophorus noctilucus), known colloquoially as the “cucuyo”, and other pyrophorine genera from the southern US (e.g., Deilelater, Ignelater, Vesperelater). The bioluminescence of P. noctilucus is so bright it can be seen from afar and, according to ship logs, was confused by Spanish conquistadors with the smoldering matches of arquebusas held by indigenous inhabitants, thereby discouraging attack (Perkins, 1868). Eyespots are often used to deter predators and function to deflect attack to a non-vital body region or to startle predators (Skelhorn et al., 2016). These functions are the false-head (“lose-little-to-save-much” ref. (Sourakov, 2013)) and deimatic strategies (Umbers, Lehtonen & Mappes, 2015); however, in the case of the deimatic function it remains uncertain whether the eyespots deter attack because they appear as eyes (often of a larger, more intimidating animal) or due to their conspicuousness (Skelhorn et al., 2016). The false-head hypothesis for click beetles with eyespots atop their pronotum seems unlikely since the thorax houses vital organs such as the dorsal vessel and thoracic ganglia. Noting the similarity between the eyespots of the cucuyo and the eyed elater, McDermott (McDermott, 1911) examined the latter to determine if the eyespots were “luminous, or at least have beneath its chitin some structure indicating that the eyespots were a degradation of the photogenic organs of the cucuyo”. Although he found thicker cuticle underlying the eyespots, potentially due to muscular attachments of the thoracic cavity, no structures consistent with the photogenic organs of the cucuyo were found. McDermott (McDermott, 1911) remarked that the false eyes may be “an extraordinary development of protective colouration.”

Figure 1 Eyed elater click beetle, Alaus oculatus, dorsal habitus and false eyes.

Eyed elater click beetle, Alaus oculatus, (A) dorsal habitus view, (B) left lateral view (scale bar = 2.0 mm); eyed elater false eyes, (C) left dorsal view, (D) right oblique view (scale bar = 0.5 mm).

As part of a study of structural coloration of insects in the Virginia Tech Insect Collection, we found striking examples of iridescence, but while observing the eyespots of A. oculatus in the collection and in the field in the Appalachian Mountains, we were struck by their profoundly black appearance at all angles. By depositing a thin metal film on the eyespots to prohibit absorption by pigments, we tested if structural absorption provides the black appearance. We compared the microstructures inside the periphery of the eyespots versus elsewhere on the exoskeleton of the beetle, and with the surface structural morphology of instances of super black in nature.

Material and Methods

We used material preserved in the Virginia Tech Insect Collection for this study (VTEC, collection.ento.vt.edu). Three adult specimens of A. oculatus were selected for the analysis. The individuals were pinned dried specimens collected from Virginia, Delaware, and Texas (U.S.A.) with the following VTEC catalog numbers: VTEC000000784, 5065, 5068, and 5069. To visually examine gross morphology, the eyespot of each specimen (right side) was examined at 45° and 90° angles with a Leica M125 stereomicroscope illuminated by a LED fiber optic light source. Setae composing the eyespot and the white ring encircling the eyespot (the “eyeliner”) were removed with a straight-edge razor and mounted in glycerin on a microscope slide. Photographs of the setae were made with a Zeiss Axio Imager A2 microscope and AxioCam ERc5s camera, and a Leica DM500 microscope. The beetle specimen was photographed with a Canon EOS 6D digital SLR camera illuminated with two Canon Speedlite 430EXII flashes diffused with a paper cylinder.

To test the hypothesis that structural absorption contributes to the deep black appearance, eyespots were plasma coated with a thin layer of platinum (Pt) and palladium (Pd) metals to control for absorption by pigments. From the middle of the right eyespot, including a piece of the eyeliner and surrounding cuticle, a 4.25 × 2.3 mm2 tile was removed with a straight-edge razor and affixed on a 12.7 mm diameter aluminum scanning electron microscope (SEM) stub with double-sided carbon tape. The stub was plasma coated under stable argon pressure with 20 nm of Pt-Pd metals with a Leica EM ACE 600 high vacuum coater, and imaged on a FEI Quanta 600 FEG environmental SEM (5 kV, 3.5 spot size). A second round of 80-nm coating was carried out to ensure that pigmentation was entirely concealed. Elemental analysis with energy dispersive X-ray spectrometry (EDS) was then used to confirm that the SEM stub was evenly coated with metals (Bruker AXS microanalysis system with XFlash SDD and e-Flash EBSD detectors). The elemental composition of two sample areas—the stub surface and the eyespot setae—were compared with EDS to test if the shape of two ED spectra were different (not superimposable) and therefore indicative of a Pt-Pd coating deficiency. The width, spacing, direction and density of setae on the eyespots were calculated from the SEM images using the program ImageJ version 1.52k (Rasband, 2019).

To measure reflectance, we used a spectrometer attached to a light source by a 400-µm diameter fiber core reflectance probe with a 24.8° acceptance angle (Ocean Optics USB 4000 spectrometer, QR400-7-UV fiber, and DH-BAL 2000 deuterium-halogen light source). A disc of polytetrafluoroethylene was used as a reflectance standard to calibrate the measurements (Ocean Optics WS-1). Reflectance measurements were made in a dark room with the probe oriented at a 45° and normal incidence and at a detection distance of three mm. Units are in percent reflectance, and are reflection factors, or empirical measurements of intensity normalized by the intensity of the reflectance standard. Reflectance was measured between 300 –700 nm, which encompasses the visible range of most animals. The eyespot, eyeliner, and the exoskeleton were measured from two individuals (VTEC000005068, 5069) three times. To compare the eyespot of the eyed elater against known instances of super black, the following butterfly specimens from the VTEC were measured using the same procedure: Ulysses swallowtail (Papilio ulysses, catalog number VTEC000000357) (Vukusic, Sambles & Lawrence, 2004), Rajah Brooke’s birdwing (Trogonoptera brookiana, VTEC000005067) (Han et al., 2015), and the common rose (Pachliopta aristolochiae, VTEC000005066) (Siddique et al., 2017). Then, to quantify the contribution of light absorption that is structural, the reflectance of the eyespot and eyeliner were measured after metal-coating to prohibit pigmentary absorption. The R package pavo was used to analyze and visualize the reflectance measurements (Maia et al., 2013). To calculate overall percent reflectance between 300–700 nm, area beneath the curve of the spectrum was summed with Riemann sums and divided by the total area of 100% reflectance between 300–700 nm. Reflectance spectra were averaged, standard error of the mean calculated, and plotted using the R package pavo.

Results

Based on visual examination with the light microscope, A. oculatus is generally black with white irregularly-shaped spots speckled across the body. The beetle possesses two large velvety black spots on the pronotum that are fringed in white eyeliner (Figs. 1C–1D). The beetle is generally clothed with V-shaped seta of varying hue and texture, and the irregularly-shaped spots, white eyeliner, and eyespots are made up of this seta. Outside of the eyespots, and generally distributed across the cuticle of the beetle, the setae are brick red and have a smooth surface. The cuticle outside of the eyespots is smooth and glossy with lustrous specular reflection. The setae of the white eyeliner are rounded apically, translucent, and lack pigment (Figs. 2A, 2B). Eyeliner setae and those outside of the eyespots are decumbent (Figs. 2F –2H). In contrast the setae inside the periphery of the eyespots are erect, acuminate apically, black with longitudinal grooves, and evenly spaced (Figs. 2A, 2C–2H, 2K, 3A). The V-shaped setae in the eyespots are more canoe-shaped than the others, with a flat slightly concave face opposite of the convex (hull) side (Figs. 2A, 2C, 2D, 2K). With the grooves, these setae appear as caraway seeds cut in half longitudinally, striped with lines running along its length (Figs. 2A, 2C–2G, 2K). By fracturing its cuticular layer, the eyespot and eyeliner setae contain numerous densely-packed melanosome-like spherules (Figs. 2I, 2J, 2L, 2M). The cuticle underlying the eyespots is glossy and similar to the cuticle outside of the eyespots; however, its surface is recessed and dimpled around setal sockets (Figs. 3B –3D).

Figure 2 Eyed elater click beetle, Alaus oculatus, setal morphology of false eyes and eyeliner.

Eyed elater click beetle, Alaus oculatus, setal morphology, (A) transmitted light color photograph of eyespot (bottom) and eyeliner (top) setae (scale bar = 50.0 µm), (B) schematic of beetle head and pronotum showing measurement areas in ‘F’ and ‘H’ (scale bar = 4.0 mm), (C) scanning electron micrograph (SEM) of the eyespot setae, magnified 326X (scale bar = 0.2 mm), (D) same, 1,247×(scale bar = 0.05 mm), (E) same, 5,033×(scale bar = 0.01 mm), (F) SEM of cephalic right corner of eyespot (scale bar = 250 µm), dashed rectangle (in G) showing perpendicular eyespot setae on left and decumbent eyeliner setae on right (scale bar = 100 µm), (H) SEM of cephalic side of the eyespot (scale bar = one mm), (I) SEM of interior of an eyespot seta with melanosome-like spherules packed inside (scale bar = 2 µm), (J) same, 100,000×(scale bar = 200 nm), (K) SEM of eyespot seta (scale bar = 60 µm), (L) SEM of interior of an eyeliner seta with melanosome-like spherules packed inside (scale bar = 2 µm), (M) same, 100,000×(scale bar = 200 nm).

Figure 3 Eyed elater click beetle, Alaus oculatus, false eyes, lateral view and exoskeletal dimples of eyespot setae.

Eyed elater click beetle, Alaus oculatus, false eyes, (A) lateral view (scale bar = 1.0 mm); (B) right dorsal view, dimples denoted by arrows, eyeliner setae in middle (scale bar = 0.5 mm) (left side of A and B is cephalic); (C) dorsal view with setae removed (scale bar = one mm); (D) same, 728×(scale bar = 100 µm).

When viewed at a 90° angle, even with the bright illumination of the microscope (illuminated at ca. 45°), little surface structure was apparent and the eyespots appeared profoundly black, appearing as voids in the body (Fig. 1C). When viewed at a 45° angle (and more acute angles), surface structure was discernable and the setae appeared regularly spaced with the concave side (hull) of the canoe-shaped hairs consistently facing outwards in a ring encircling the periphery of the eyespot (Fig. 3A, Fig. S4). The other half of the setae are in the center of the eyespot and appear to be facing random directions (Fig. S4). Along the periphery, the setae of the eyespots are bent at a ca. 60° angle and gradually change in angle to 90° at the center of the eyespot; in contrast, the setae at the posterior-facing margin remain upright and ca. 90° (Fig. 3A, Fig. S1).

Figure 4 Eyed elater click beetle false eyespot and eyeliner coated with a 100-nm layer of platinum and palladium.

Eyed elater click beetle, Alaus oculatus, eyespot (top white arrows) and eyeliner (bottom black arrows) coated with a 100-nm layer of platinum and palladium, (A) 15° view, (B) 30° view, (C) 45° view, (D) 60° view, (E) 75° view, (F) 90° view (scale bar = 1.0 mm).

The EDS analysis of the sample areas resulted in ED spectra with the same shape (superimposable spectral profiles) and identical elemental composition with uniform coating (Figs. S1–S3). Micrographs of the eyespot setae at 2.5 ×104 magnification were recorded without building of charges indicating a fully conductive surface and a complete Pt-Pd coating (Figs. S4). As a result of plasma coating, the SEM stub and non-eyespot portions of the cuticle sample possessed a lustrous metallic surface. When viewed from the side at a shallow angle (15–30° from the surface of the stub), the eyespot setae have a specular metallic sheen, indicating thorough coating. However, despite uniform metal coating, the eyespot retained a deep black appearance at normal incidence (Fig. 4, Movie S1). Based on examination of the eyespots with the SEM (between 326–5,033×magnification), the V-shaped seta are lined with about 14 longitudinal ridges (mean = 13.97, standard deviation = 1.92, n = 30) with a smooth somewhat concave opposing surface (Figs. 2A, 2D, 2E). The ridges that line the setae longitudinally are each 4.28 µm wide (mean, standard deviation = 0.45 µm, n = 30). The pointed apices of the setae are dividing into two or occasionally three shallow furcations (Fig. 2E). Setae are about 38.33 µm in widest width (mean, standard deviation = 2.15, n = 30) and spaced about 10.30 µm edge-to-edge from one another (mean, standard deviation = 2.67, n = 30). Within in a 726.68 µm2 area of the eyespot there are 250 setae, and 1,756 setae within the eyespot area in total.

From the measurement of reflectance, the white eyeliner, the glossy exoskeleton and the black eyespot possessed spectra of different shapes (Fig. 5). The black eyespot spectrum was a flat profile, indicating a general lack of reflectance across a broad range of wavelengths. The white eyeliner spectrum had a plateau shape, lacked near-ultraviolet reflectance, and with uniformly high reflectance between 400–700 nm. The glossy exoskeleton spectrum was generally flat in profile but with consistently high overall reflectance (including ultraviolet) indicating a high glare from the lustrous cuticular surface. The overall reflectance of the black eyespot patch was 3.90%, the white eyeliner was 32.92%, and glossy exoskeleton was 46.14%. The overall reflectance of the super black wing patches of the butterflies T. brookiana, P. ulysses, and P. aristolochiae were more than six-fold less than that of the beetle’s eyespot: 0.51%, 0.55%, and 0.71%. The overall reflectance of the black eyespot measured at a 45° specular orientation from three directions (posteriad, mesad, and anteriad) were ten-fold less than that at 90° : 0.30%, 0.36%, and 0.38%. To quantify the contribution of light absorption that is structural, the overall reflectance of the metal-coated structures were 10.38% for the eyespot (2.66-fold greater than the uncoated) and 44.56% for the eyeliner (11.42-fold greater than the uncoated).

Figure 5 Reflectance spectra of the eyed elater eyespot and butterflies with super black wing scales (A), and schematic (B) of proposed structural light absorbing mechanism (solid line = light propagating through air; dashed line = light propagating through solid), blue spectrum line = eyespot (ey), red = white eyeliner (el), green = glossy cuticle (cu), dashed brown = Ulysses swallowtail (pictured), dashed purple = Rajah Brooke’s birdwing, dashed teal = common rose.

Discussion

We found that structural absorption gives a black appearance to the eyespots of A. oculatus. By depositing a thin metal film on the eyespots to prohibit light absorption by pigments, we demonstrated that pigmentation alone is not responsible for their deep black color. Based on our examination of the black eyespots, we found that their surface morphology was equivalent in shape, orientation, and general photonic properties of super black structures in nature. In particular, the eyespot composed an array of perpendicularly aligned linear protuberances that absorb 96.1% of light and is analogous to the three-dimensional array of microtubules on butterfly wings (Vukusic, Sambles & Lawrence, 2004) and the ramified barbules on bird of paradise feathers (McCoy et al., 2018) and jumping spiders (McCoy et al., 2019). However, the perpendicularly aligned array of setae on the eyed elater’s eyespot is scaled up ten-fold relative to the super black examples in other animals (10 µm between neighboring eyespot setae versus 1 µm in spiders). Though the morphology is analogous, the magnitude of scale may account for the quantitative difference in absorption of light. Other examples of structural black in nature with similar perpendicularly aligned protuberances include the cuticular papillae of stick insects (Maurer, Kohl & Gebhardt, 2017) and leaf-like microstructures on viper and peacock spider scales (McCoy et al., 2019; Spinner et al., 2013). Functionally equivalent super black structures in butterflies include polydisperse honeycomb-like meshes of the wing scales of the Ulysses swallowtail (Vukusic, Sambles & Lawrence, 2004), Rajah Brooke’s birdwing (Han et al., 2015), and the common rose (Siddique et al., 2017). Though a different morphology, these meshes are ostensibly negative casts of perpendicularly aligned microtubule arrays, and synthesized SiO2 negative replicas derived similar light-absorbing capabilities of their biological templates (Han et al., 2015). All of these surface structures generate multiple reflections and refractions causing light to travel a greater distance, causing structural absorption and leaving little light available to reach the viewer’s eye. While human-fabricated super black materials typically absorb 99.965–99.995% of light (Cui & Wardle, 2019; Theocharous et al., 2014), in nature, bird of paradise feathers come close to this with incident reflectance of about 0.05%. Butterflies and spiders with super black patches reflect more (incident) light with between about 0.35%–0.71%.

Melanin is a ubiquitous black pigment of insect exoskeletons, and is a component of the setae of the eyed elater’s eyespots. Melanin contributes to the black color of the eyespots by absorbing light in concert with structural absorption. The pigment directly absorbs light and perhaps also recaptures light that strays from structural absorption. We showed that the eyespot setae sit in a concavity (Figs. 3B–3D), and the cuticle underlying the eyespots has a dimpled topography. These concavities are smooth and have a black pigmentation. Other arthropods possess similarly shaped concavities that scatter light and impart additive mixing (of blue and yellow iridescence such as in the emerald swallowtail (Vukusic, Sambles & Lawrence, 2000)), or augment melanin absorption thereby increasing overall absorption and producing super black (as in peacock spiders (McCoy et al., 2019)). These concavities of the eyed elater’s eyespots and the fourteen 4.28-µm wide longitudinal ribs on the setae could be features that assist to impart a black color. Additionally, scattering of light may be directional (e.g., backwards) given the shape and orientation of the setae on the eyespots. Optical modeling integrating these features would be fruitful to understand how this ensemble of features work in concert to affect light.

The function of super black eyespots in the eyed elater may be for predator deterrence including aposematism, deimatism, or as a false head. A role as a false head is unlikely since the eyespots are in close proximity to the real head and not posteriorly located as in other insects (e.g., hairstreak butterflies). Since there are large dorsal intersegmental muscles directly beneath the eyespots (Evans, 1973), thermoregulation or muscle-heating is another functional hypothesis. While the startle function is the most likely, click beetles do have a powerful clicking mechanism that is noxious to birds (Evans, 1973; Eisner, Eisner & Siegler, 2005) and disentangling this aposematic role versus a startle function would ideally be tested using field experiments.

The study of super black structures in nature have uncovered a diversity of morphologies that cause near complete absorption of light. A result of convergent evolution, analogous structures such as honeycomb-like meshes and perpendicularly-aligned arrays have originated to act as a general baffle of light for various functional roles. Since super black materials have application for human industry (e.g., solar cells, artistic expression, etc.) the structural morphology of these various materials in nature are an ideal domain as creativeness for fabricating structures and as a means to understand the evolution of adaptive coloration and natural selection.

Conclusions

In this study, we asked: what makes the eyespots of the eyed elater black? We found that the eyespots comprise an array of perpendicularly aligned setae with black pigmentation. The eyespot is circled by a ring of clear setae that lay flat and appear white, the “eyeliner”. The black eyespot absorbs 96.1% of incident light, ten-fold more than the eyeliner. A collaboration between structural and pigmentary absorption, multiple reflections and refractions increase the distance that light travels thereby reducing the amount of light available to the receiver. The intense black of the eyespot provides a stark contrast versus the eyeliner making the eyespots conspicuous to a predator. Ostensibly an aposematic signal to warn of the beetle’s noxious clicking behavior and increased handling time, the highly apparent eyespots may also serve a startle (deimatic) function making the beetle appear as a larger, more formidable opponent.

Supplemental Information

Supplemental Information 1 Supplementary figures

Figure S1, graphical denotation of the rotation of the setae within the eyespot. Supplementary Figure S2, energy dispersive X-ray spectrometry (EDS) sample areas. Figure S3, EDS spectra of sample areas. Figure S4, scanning electron micrograph of the bases of eyespot setae that shows magnified view (25,000×magnification) without building of charge indicating thorough sputter coating of platinum and palladium metals.

Click here for additional data file.

Supplemental Information 2 Supplemental Movie: eyed elater click beetle, Alaus oculatus, eyespot coated with a 100-nm layer of platinum and palladium, tilt sequence

Eyed elater click beetle, Alaus oculatus, eyespot (top white arrows) and eyeliner (bottom black arrows) coated with a 100-nm layer of platinum and palladium, (A) 15° view, (B) 30° view, (C) 45° view, (D) 60° view, (E) 75° view, (F) 90° view (scale bar = 1.0 mm)

Click here for additional data file.

Supplemental Information 3 Spectrum files and R script

42 text files with reflectance spectra and 1 R script text file

Click here for additional data file.

Thanks to Drs. Ellen Brown and Barry Lee Bressler for support of the Virginia Tech Insect Collection and a donation of beetle specimens, including the two eyed elaters used in this study. Jackson Means assisted with counting the ridges on setae. Doro Tholl provided access to the Zeiss microscope. Steve McCartney and Chris Winkler at the Nanoscale Characterization and Fabrication Laboratory at the Virginia Tech Institute for Critical Technology and Applied Science assisted with SEM and EDS. We are grateful to Charity Hall for editing previous versions of the manuscript, and anonymous reviewers for suggestions and improvements.

Additional Information and Declarations

Competing Interests

Author Contributions

Data Availability

The authors declare there are no competing interests.

Victoria L. Wong conceived and designed the experiments, performed the experiments, analyzed the data, contributed reagents/materials/analysis tools, authored or reviewed drafts of the paper, approved the final draft.

Paul E. Marek conceived and designed the experiments, performed the experiments, analyzed the data, contributed reagents/materials/analysis tools, prepared figures and/or tables, authored or reviewed drafts of the paper, approved the final draft.

The following information was supplied regarding data availability:

The raw data are available at: VTechData: https://data.lib.vt.edu/collections/kp78gg534, DOI 10.7294/G2QQ-0K42.

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
