# Peer review of "Structure and pigment make the eyed elater’s eyespots black"

_PeerJ, doi:10.7717/peerj.8161_

## Round 0.1 · original submission · Major Revisions

Please carefully review the two critical reviewers comments/suggestions (R2 & R3) and address each in your resubmission. Pay particular attention to the issue of the definition used of super black, as your materials are not within the absorbance range typically referred to as super black. Also of critical importance is the issue of providing 1. comparative evidence for the non-black scales, 2. higher magnification images of the ultrastructure, and 3. some way of confirming that the pigments have been eliminated from the absorbance tested.

·

Basic reporting

The information reported here is interesting, valuable, and relevant to the literature on beetle structural colors. I was surprised this had not been published on before, and am glad Alaus is receiving the attention it deserves.

The only unprofessional English usage is a sprinkling of arbitrarily capitalized common names, which I have highlighted in the attached Word file.

Experimental design

The experimental design here is more than adequate to demonstrate that the "super-black" of the eyespots arises from a structural mechanism. There are a few small observations that could be added (SEM investigation of the interior structure of the scales; spectrometer measurements from additional directions relative to the beetle's longitudinal body axis) but these are by no means essential.

This research fills knowledge gaps around "super-black" structures in Coleoptera AND surrounding eyespots in Coleoptera, which have received little or no synthetic attention in the literature (to my knowledge, there are a few instances documented from tiger beetles).

Validity of the findings

These findings are valid and appear robust and reproducible. Statistics are not a concern here. Conclusions and discussion are well-linked to previous research.

Additional comments

Please see attached Word file for additional notes and markup.
Overall, this is a fascinating paper and will be a valuable contribution to the literature on beetle structural colors.

If it's possible to break open some of the scales (I suggest macerating them a bit with a pin or razor blade against a glass slide) and SEM their interior or cross-section, this may also shed light on how the contrasting light/dark of the eyeliner/eyespots is achieved. However, this manuscript is interesting enough on its own and I don't think it requires additional experimentation (and much setal morphology is already revealed through the combination of external SEM and transmission light microscopy).

Reviewer 2 ·

Basic reporting

The authors reported the natural “super black” structures in the false eyespots of the eyed elater. In fact, a few cases on natural “super black” have already been reported involving butterfly wings, bird feathers, moths, etc. The structure investigation on the black eyespots of Alaus oculatus is interesting, which enriches the structure database of natural “super black”. However, based on my careful assessment on the overall work, the manuscript at current stage is not suitable for publication in PeerJ before the authors can well handle the following issues:

Experimental design

1. The author demonstrated the intuitional observation results of the “super black” false eyespots of Alaus oculatus including digital photographs and SEM images. Apparently, these data are necessary but not enough to form a relatively complete research article. The underlying optical mechanism of the “super black” in this case should be discussed thoroughly and a set of schematic diagrams could be very helpful to make it clear for a wide audience.
2. To help distinguish the difference as compared to other natural “super black” structures, the feature structure details of the eyespot setae should be highlighted in Figure 2.
3. For reflectance spectra in Figure 5, to exclude the impact of melanin to the “super black”, a comparison group of samples should be included. One group is the original sample without post-treatment and another group should be the treated sample with thin metal coatings.
4. Optical modeling of the “super black” structures should be built up and corresponding optical simulation involving the structure models and incident light using FDTD or other recognized methods should be provided to improve the reliability of the conclusion.

Validity of the findings

No comment.

Additional comments

1. “Super black” should be clearly defined in this manuscript and point out the difference between “super black” and ordinary black.
2. The authors claimed the structure similarity of natural “super black” between Alaus oculatus and other animals based on their shape, orientation, and general photonic properties. However, as I known, their feature structures responsible for “super black” are quite different. Thus, the structure differences of these typical biological prototypes of “super black” should be compared in details.
3. Other small issues were also highlighted as notes in .pdf version of the manuscript, please double check.

Annotated reviews are not available for download in order to protect the identity of reviewers who chose to remain anonymous.

Reviewer 3 ·

Basic reporting

The article is well-written, and most of the appropriate literature references are included. I suggested some additional papers to include. The figures look nice and professional. I think a comparison SEM between black and white setae is needed since that seems to the be the main point of the paper, and higher magnification images are needed as well.

Experimental design

The research question is well-defined, but the data shown in the paper do not clearly answer the hypothesis that the eyespots are structural black.

Validity of the findings

Since only low magnification SEMs are provided, and there is not a comparison SEM of the white setae, it's currently impossible to fully assess the validity of the findings. A previous reviewer suggested that these authors sputter coat the sample to show if it's structural black, but this sputter coat experiment is not enough to confirm structural black. Suggestions follow in the detailed pdf attachment about how the authors may confirm structural black.

Additional comments

See attached pdf for these same comments but perhaps formatted better:

The manuscript “Super black eyespots of the eyed elater” describes how the “super black” false eyespots on the surface of a click beetle may be as black as they are due to structural absorption of light as well as pigmentary absorption. While this manuscript is well-written and gives sufficient background and context with references to other optical structures found in the “super black” literature, there are several major issues that need to be addressed before it would be suitable for publication. Most importantly, the authors claim to test if structural absorption provides the super black appearance, but the data shown here (1) do not support that the false eyespots are even a “super black” structure to begin with given that they only absorb 96.1% of incident light, which is more equivalent to black construction paper than a “super black” material or animal (black butterflies absorb 99.95% of light) and (2) do not support that the “perpendicularly aligned protuberances,” or setae, are the reason that the false eyespots are black. The experiment to sputter coat the eyespot with platinum-palladium was interesting but this still doesn’t confirm that the black is structural. Sputter coating doesn’t always penetrate deep in between microstructures like we would hope and it looks like the pigment at the bottom of the setae simply isn’t covered up. The authors may try bleaching the spot to confirm. There is also an issue of relating morphological structures that are on size scales that differ by orders of magnitude. For example, the nanoprotuberances on the eyes of moths and the polydisperse honeycomb structure acting as an aggregation of resonant absorbers that are found on many black butterflies are not on the same size scale as the setae the authors describe here on the black eyespots of their beetles. The authors do not do any optical modeling, and while I agree that an optical model is not necessary for publication of their manuscript, working through some of the optics and relating the morphological structure to the reflectance measurements might help the authors understand why their setae is not equivalent to other super black nanostructures that have been previously described in the literature. I suggest the authors reframe the point they are trying to make in the study since the spots aren’t even “super black.” However, it is interesting to use SEM to characterize the cuticular surface of the click beetle, as the authors begin to do in this study, but the measurement needs to be a direct SEM comparison between the black spots and the surrounding white. The authors provide a light micrograph of the white setae, but never even provide an equivalent SEM of the white part. If the authors have these data that support that the white setae are structurally different in the nanoscale than the black setae, they need to add a figure or show that additional data somewhere in the paper.



Technical details and line edits:


Introduction
Line 57 – reference 11. Corneal nipple arrays function quite differently than other structural black examples and wouldn’t actually be classified as a “super black” structure just an antireflective structure.

Line 64 – antireflective coatings are everywhere, not just use in the military. Example: eyeglasses, window panes, etc. Also, this is a different concept than super black

Line 97 – by depositing the sputter coating on the surface, you did not unequivocally confirm that you had covered up the pigment. If you zoom in enough on the SEM, you should be able to see if the sputter coating actually reached and covered the base of each of the setae. If you can’t zoom in/magnify to that level, you are probably getting charging of the surface and thus you didn’t actually coat it like you thought you did.

Line 98 – you say you compared microstructures from the black spots with areas elsewhere on the cuticle, but you never show direct SEM comparisons here. I only see the figure that shows the difference at the light micrograph level.


Materials and Methods
Line 125 – how do you confirm that the sputter coating was 40nm (20 nm + 20nm)? If it’s like most sputter coaters, you enter this thickness, or you sputter for a certain amount of time previously known to achieve that thickness. However, if you magnify to the point where you see structures at the size scale of 1 – 10nm, you could actually see the sputtering on the surface – did you do this then to confirm it actually covered the setae all the way to the base? The SEMs provided in Figure 2 are not very magnified at all (highest magnification has a scale bar of 0.01mm so it’s impossible to tell).

Line 132 – for most “super black” materials, a white reflectance standard (whether Spectralon or other known calibration) is usually too bright and then results in nothing but noise when measuring reflectance of the “super black” material. However, the beetle false eye spots aren’t even that black, so you may not need to use a black spectralon standard for your reflectance measurements. Still, I’d try it with a different standard to see if that affects your reflectance measurements.



.
Results
Line 168 – why don’t you show any SEMs for the white setae? Figure 2A shows a transmitted light photograph, but we need to see a direct comparison SEM. It looks like the white setae are longer and that is why they are lying down, but even from the text, it’s unclear what is happening at the nanoscale with both the white and black setae.

Figure 3 and Figure 4 – again, the previous reviewers had a good suggestion to look to see if sputter coating would cover up the black pigment. But it looks like the black setae are very dense and it’s very difficult to sputter all the way to the base of dense structures. The supplemental video and the photographs at 30 degress and 45 degrees show the tips look coated, but it’s impossible from just the data shown to confirm it coated to the base. Try bleaching the pigment away.


Discussion:
Line 222 – The sputter coating was a good idea to start with, but this alone doesn’t confirm that it’s structural black just because it still looked black looking directly down onto the spot.

Line 224 – The large setae are “erect, black with longitudinal grooves, and evenly spaced” (quoted from line 170), but this is not “equivalent in shape, orientation, and general photonic properties of other super black structures.” You have not magnified the setae enough in any of the provided SEMs to show the nanostructure arrangement at all. If you have these data, please add them to the manuscript.

Line 230 – the nipple arrays of moth eyes are not at all similar to the SEMs you show in figure 2 in either scale or order. They are perpendicularly aligned, but lots of structures in biology are that have no optical function. This is where some discussion of the optics behind what makes structural black would be useful. You don’t need to make your own optical model for this paper, but you need to recognize the difference in scattering that would occur between a 20nm structure and a 2000nm structure.

Line 235 – is the 0.5% a typo? The blackest of which are made from a forest of carbon nanotubes are capable of reflecting as little as 0.035% of incident light (Theocharous, E., Chunnilall, C.J., Mole, R., Gibbs, D., Fox, N., Shang, N., Howlett, G., Jensen, B., Taylor, R., Reveles, J.R., Harris, O.B. 2014. The partial space qualification of a vertically aligned carbon nanotube coating on aluminium substrates for EO applications. Opt.s Exp. 22 7290-7307). And animals have evolved micro- or nanostructures that reflect as little as 0.05% of visible light. So I don’t think your citation of the birds of paradise at 0.5% is correct. Black things can reflect 3-11% but they don’t often fall in the “super black” category.


References
Also check out the following:
1. Han, Z., Li, B., Mu, Z., Yang, M., Niu, S., Zhang, J., Ren, L. An ingenious super light trapping surface templated from butterfly wing scales. Nano. Res. Lett. 10, 344-351(2015).
2. Siddique, R.H., Donie, Y.J., Gomard, G., Yalamanchili, S., Merdzhanova, T., Lemmer, U., Hölscher, H. Bioinspired phase-separated disordered nanostructures for thin photovoltaic absorbers. Sci. Adv. 3, 1700232-1-1700232-11 (2017).
3. Theocharous, E., Chunnilall, C.J., Mole, R., Gibbs, D., Fox, N., Shang, N., Howlett, G., Jensen, B., Taylor, R., Reveles, J.R., Harris, O.B. 2014. The partial space qualification of a vertically aligned carbon nanotube coating on aluminium substrates for EO applications. Opt.s Exp. 22 7290-7307

Annotated reviews are not available for download in order to protect the identity of reviewers who chose to remain anonymous.
External reviews were received for this submission. These reviews were used by the Editor when they made their decision, and can be downloaded below.

---

## Round 0.2 · accepted · Accept

Please attend to the suggestions for minor revision pointed out by the reviewer prior to working with PeerJ production staff to prepare the manuscript's final draft. These suggestions will indeed improve the manuscript.

Reviewer 3 ·

Basic reporting

Authors addressed my previous concerns with lit references.

Experimental design

Authors addressed my previous concerns with defining the research question.

Validity of the findings

Authors addressed my previous concerns with the conclusions.

Additional comments

The authors have done a good job of addressing my previous comments, but there are a few minor issues to still be addressed prior to publication, mainly having to do with Figure 2 needing further clarification. Please see below for line edits.

Abstract, line 22: edit the sentence to say something like -
'Here we describe the false eyes of the eyed elater click beetle, which, while not classified as "super black," still attains near complete absorption of light partly due to an array of vertically aligned microtubules.'
Somewhere it needs to be clear that it's not super black, but you're going to talk about how it is black due to structure as well.

Figure 2: needs more clarification
B - I would suggest getting rid of the entire panel B. It seems to show the same thing as one of the eyeliner setae in panel A, so it is just confusing to have a different picture of it in a separate panel.

F, G, H - it is not clear exactly what is being shown here. I think you are trying to show the vertical eyespot setae next to the decumbent eyeliner setae, but I can't tell where the transition happens in panel F. Please add arrows to clarify this, or find some other way to indicate which is eyeliner and which is eyespot.

Can you add a photo of the beetle (or section of the beetle) and then add a dashed rectangle showing the region of interest and relate that to where F, G, and H are on the beetle? That would be good to do for all panels. You use the rectangle section idea effectively when going from panel I to panel J. I think you need to go back one more step and show where everything is on the beetle. Having a bunch of SEMs all arranged like that makes it hard to understand, even when you're carefully reading the caption. It wasn't immediately clear to me that you addressed my previous concern about not having the eyeliner setae SEMs.